# Does Sociology Need Open Science?

**Nate Breznau** 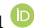

Department of Sociology, SOCIUM Research Center on Inequality and Social Policy, University of Bremen, 28359 Bremen, Germany; breznau.nate@gmail.com

**Abstract:** Reliability, transparency, and ethical crises pushed many social science disciplines toward dramatic changes, in particular psychology and more recently political science. This paper discusses why sociology should also change. It reviews sociology as a discipline through the lens of current practices, definitions of sociology, positions of sociological associations, and a brief consideration of the arguments of three highly influential yet epistemologically diverse sociologists: Weber, Merton, and Habermas. It is a general overview for students and sociologists to quickly familiarize themselves with the state of sociology or explore the idea of open science and its relevance to their discipline.

**Keywords:** open science; crisis of science; sociology legitimation; transparency; science community; p-hacking; publication bias; replication; research ethics; Merton; Weber; Habermas

## 1. Open Science: Some Housekeeping

This paper is motivated by the open science movement, which is a loose array of different ideas and practices with the central claim that science in practice is problematic. This paper assumes basic awareness of the concept of open science; if not, some definitions and resources are provided in Appendix D. Open science ideas are not new. Scholars long ago warned of the problems with academic status competition, perverse institutional incentives, inter-subjectivity issues, and a lack of transparency. In fact, sociologists Weber and Merton were pioneers in addressing these issues in the early and mid-20th Century. What is 'new' today is overwhelming empirical evidence that these issues have not been dealt with. Moreover, there is strong evidence that the public and policymakers have low trust in and respect for science, social sciences in particular. Thus, it is my position that sociology needs open science.

The events and facts presented herein do not constitute new knowledge. These are collected here because the extensiveness of the need for open science is likely unknown to most sociologists. It is therefore a compilation of evidence. The paper argues that sociology needs changes, and simply uses the term "open science" to describe them. As such, readers could extract the same information from this paper without the term "open science", the message would remain. Regardless of label, the paper raises awareness of serious problems that are rarely discussed in mainstream sociology. I encourage the reader to consider these problems, independent of their own perceptions of the term and the movement. If sociology is to change as suggested, this change must come from academic and research institutions; local, national, and international politics; publishers; and researchers themselves. This paper should motivate such changes. If researchers want to know how to practice open science now, they will find some ideas in Appendix E.

This paper is not authoritative. It is not a scientific study. It is an excursus on the state of sociology from the perspective of one sociologist. It is meant to inform or empower those who might find themselves discussing the state of their discipline or encounter others pushing open science. The paper is a success if it helps both supporters and critics of open science in sociology to better understand their own positions and to further consider these issues with their peers and in the classroom.

## 2. The Crisis in Science. The Crisis in Sociology

Scientists of all sorts indicate there is a crisis in science. This 'crisis' is that science is less reliable, reproducible, and ethical than policymakers, the public, and other scientists expected or previously believed. Widespread failures to replicate previous research or even simply obtain others' replication materials, a publishing industry often at odds with the goals of scientific research, lack of awareness of the biases in standard research practices, and individual researchers who become ethically corrupted in seeking recognition and funding are a few of the reasons for this crisis. Whatever the label, and regardless of how long it has been ongoing, I argue it is an ideal time to take stock of sociology. I construct arguments herein suggesting sociology is no exception to these problems. If so, sociology in practice is quite far from sociology by ideal definition.

Scandals that brought social science into the limelight mostly occurred in disciplines other than sociology. Some important events: In psychology, Diederik Stapel spent a career faking data and results that were published in at least 54 articles that consumed millions of Euros in funding until several whistleblowers outed him (more details available in the 2012 final report of the joint Levelt, Noort, and Drenth committees); in political science, LaCour and Green published a study in *Science* that attitudes toward gay marriage could be changed if heterosexual people listened to a homosexual person's story, but it turns out LaCour fabricated results for a 'follow up survey' that never took place as uncovered by Broockman [1]; and in economics, Brian Wansink "misreported" (as in faked) data leading to 18 retractions and Reinhart and Rogoff published studies identifying a negative impact of high debt rates on national economic growth, when in fact several points in their dataset had conspicuously missing values that when corrected took away all support for their claim as identified by Herndon, Ash, and Pollin [2,3].

Could it be that sociologists are more scientific and ethical in their research behaviors than members of neighboring disciplines such as the more extreme cases listed above? I believe that because they face almost identical institutional and career structures that favor productivity and innovation over replication and sound research practices that otherwise often lead to null results, it seems unlikely [4,5]. Consider for example that sociology journals and their editors, rarely retract articles despite evidence of serious methodological mistakes [6–8]. At the same time, sociology journals rarely publish replications meaning there is little incentive for methodological critique or disciplinary self-correction.

Carina Mood once pointed out false interpretations of odds-ratios in some *American Sociological Review* articles, but the editors refused to publish her comments, much less consider retractions. She shared her exchange with *ASR* in an email to me and discusses some of it in a working paper [9]. This represents a clear lack of structural incentives for replication. An exceptional recent event was the retraction of one of Legewie's sociological studies, but this required he himself to initiate the retraction after someone pointed out errors in his analysis [10]. Until 2020, the Retraction Watch database (www.retractiondatabase.org) listed no retractions from the top sociology journals, and only two among the well-known, one in *Sociology* and another in *Social Indicators Research*; whereas psychology has dozens, not just those from Stapel. It is therefore possible that psychology is particularly unethical and unscientific as pointed out repeatedly in the replication and meta-science work of Ulrich Schimmack (see https://replicationindex.com/).

I suspect this is only evidence that sociology is less 'advanced' in the timeline of identifying questionable research practices than other disciplines. Given that a career in sociology is similar to any other science in the necessity to publish or perish (see Appendix D), I assume questionable if not occasionally fraudulent research practices take place but are not identified. In fact, while writing this paper, an in-depth investigation by Pickett [11] using replication and other techniques of assessing faked data outed Eric Stewart, sociology's first serial data and results faker that we know of. Thanks to Pickett, Stewart and sociology gained five retractions. Perhaps we sociologists should be relieved, as this is just confirmation that we are as much a part of social science and its problems as any other discipline. It is also a reminder that faked data are more likely to be discovered

when put under intensive scrutiny, and that having more replications using techniques following Pickett's would reduce practices of faking data; especially if high ranking journals were willing to publish them. As sociologists most often work with survey data, interviews, and participant observation, the opportunities to alter or even fake data are plenty. The researcher is the vessel that turns observation into evidence regardless of the origin of that evidence. Intentional fraud may be more of an outlier, but questionable research practices done without researchers' own knowledge of the consequences or intention to do any harm are likely far more prevalent. For example, a qualitative interviewer who selects interviewees who are likely to support a given hypothesis, or a quantitative researcher who runs models until finding a desired effect.

Although sociology produced many public goods throughout history [12], the institution of publish-or-perish leads to reiteration, rewording, or recycling claims that appear as new findings; a problem that sociologist turnover and the vastness of global sociology make difficult to deal with [13,14]. Whether or not they do sociology as a *means* to investigate and rigorously understand social problems, sociologists, like other scientists, do sociology as a career and this turns it into an *ends* to get a tenured position in a university or a research institute. This institutionalized nature of sociology is a basis for being skeptical of scientists including sociologists [15]. This skepticism echoes in public opinion and media messages. For example, a 2018 study of Germans found that only 26% thought scientists *did not* adjust their results to match their expectations [16]; a senior congressman in the US recently referred to social science as "sociological gobbledygook" [17]; the Wall Street Journal used the heading "fake news comes to academia" and published an opinion-editorial questioning the credibility of sociology and the social sciences [18,19]; and in a US survey, only 20% of Americans believed that scientists *always* acted in the public interest [20]. Of course, scientists have their own 'interests', but if the public believe they are not acting in the public interest, it is a credibility problem.

### 2.1. When Evidence Is Not Evident

I was taught early on that publications are 'the currency of our trade'. In the mid-20th century, some kind of 'regression revolution' took place, after which the publication-worthiness of sociological research using quantitative analyses was judged mostly by whether it arrived at a statistically significant (partial) correlation coefficient. *p*-value cutoffs became synonymous with 'important findings'. Thus, to publish an important finding, I either needed to find a significant effect and write a research design to predict this effect in the first place, or find a significant effect that supports my initial predictions. This is the simplified version of the formula: find a significant effect. On the surface, playing with data is indistinguishable from scientific research, which generally entails looking at results, refining theory, and then re-running analyses in an interplay of induction and deduction [21]. However, beneath the surface this leads to serious problems. If researchers consider hundreds or thousands of ways of analyzing or interpreting their data, they are essentially testing 'every' possible theory. When they chose only one version to report from the thousands, but pretend they had a theory predicting this unique result all along, it appears as though they found evidence in favor of that theory. The reality is that they have either selected a theory that fit certain findings or selected certain findings to fit a theory, neither of which is evidence in favor of any theory over another. In other words, it leads to unreliable and useless output touted as 'science'.

This problem is not limited to quantitative research. Sociologists using qualitative methods can report only those parts of texts, interviews, and qualitative observations that support the theory they want or the interpretations they prefer. They could consider thousands of potential subjects to observe, and only take those that might offer them what they want to find. Again, on the surface this approach could be considered sound research practice if extreme or deviant cases are the object of study. It is only when such cases are presented as common or generalizable, or when they are selected out of the intended sample contrary to a research design, that a problem of inference creeps in. It

is also problematic if findings about these or other cases are presented as results of a planned test of something, when in fact the methods were simply grounded theory. These examples reflect behaviors that lead to reporting and promoting a highly selective set of rare associations among data that correspond little, if at all, to our social worlds.

Although my own insider view as a sociologist provides only anecdotal evidence, a more convincing case comes out of ethnomethodological research conducted by Lindsay, Boghossian, and Pluckrose. They wrote a series of 20 papers that presented fake data, methods, and results. They invented the papers to mimic the style of articles published in journals well-known for sociological research on topics of identity, hegemony, and marginalization [22]. By trying to publish in these areas they were studying what they perceived to be "grievance studies", something they felt were often un-scientific. Seven of their papers were published or had revise and resubmit recommendations before a whistleblower brought an end to the project. The details are harsh, for example one paper contained sections copied from Hitler's *Mein Kampf*. Another suggested that men should be trained as canines to prevent rape, and a third that white men should be forced to sit in chains on the floors of university classrooms, instead of normal desks. I am not commenting on these ideas, any idea has potential merit, especially when it offers to correct injustices. What is instead problematic is that they all contained faked data, non-existent methods, or conclusions not supported by the data – as in they lacked any evidence whatsoever. That these studies easily flew under the radar of a number of high-impact journals raises skepticism about the reliability of sociology, if not social sciences in general.

An anonymous reviewer of this manuscript suggested that the journals targeted by Lindsay et al., are not "sociology" journals, thus implying that their research is not "sociology". This line of argumentation is problematic and suggests a definition of sociology that bars topical diversity; a sociology that only exists in general interest or mainstream journals. The authors publishing in several of the journals that were targeted, such as *Sexuality and Culture*; *Fat Studies: An Interdisciplinary Journal of Body Weight and Society*; *Sex Roles*; and *Gender, Place and Culture*, are sociologists, or working in interdisciplinary departments that include sociology; and I have no doubt that most take their sociological work seriously. They must, for they have to pass strict tenure guidelines, compete for limited funding, and train future sociologists. The journals themselves list social, behavioral, and geographic research in their descriptions. To delineate their work as not sociology is not very "open" because it would construct a disciplinary wall excluding a huge population of sociologists working on studying "collective behavior" and "social problems" related to race, class, gender, marginalization, discrimination, and identity (quotation marks added to reference a definition of sociology presented in Section 3.1).

The Lindsay, Boghossian, and Pluckrose research is a reminder that peer reviewer and journal editing systems are part of the causes of the reliability problem. Not because reviewers and editors do not do a good job, but because it is too great a burden to hold three or so somewhat randomly available and time-constrained reviewers responsible for ensuring that a paper is reliable. Regardless of whether sociologists challenge white-heterosexual-male power structures, like the aforementioned "grievance studies", or investigate any other topic: If studies are not based on real data or cannot provide reliable evidence of their claims, there is no reason to trust their conclusions or recommendations. Even in interpretivist research involving assigning motives or features to persons, events, or groups, logical arguments should correspond to evidence and vice-versa depending on the method. The reviewers found the logic compelling in these fake studies, but none questioned the data and methods. Without both, the reliability problem persists. With low reliability, the chances for sociology to impact global problems or simply secure much needed institutional resources are slim.

Several surveys of researchers, ethical integrity case files, and findings in journals in other social science disciplines reveal that up to 4% of researchers reported faking data or results at some point in their careers, that they routinely engaged in selective reporting of results, and that journals' publications contain upwards of 90% hypothesis support

rates [23,24]. I have no reason to believe sociology is any different, and if so, it is not a stretch to say that sociology faces a crisis.

### 2.2. Misinformation, Bias, and Hacking

Like all social science disciplines, sociology contains a large amount of quantitative research. The *p*-value is ubiquitous in this research, but many sociologists are misinformed about its meaning. In a randomized experiment, researchers try to claim counterfactually, what would have happened if a treatment was not administered to a group of people. The *p*-value provides information that supports or opposes such a claim. Sociologists rarely conduct experiments, but during the 'regression revolution' they began to apply *p*-value logic to their non-experimental research. The *p*-value became the gold standard, and the results of studies are now often judged entirely on whether they have a *p*-value below a certain threshold or not—*p*-value here is synonymous with a *t*-test and its equivalents; a t-value of 1.96 or a *p*-value of 0.05 carries the same gate-keeping meaning. It is problematic to judge studies in this way, because the *p*-value is only useful if the data-generating model is correct. In experiments, random assignment of participants makes the data-generating model correct in theory, and the *p*-value then implies how likely the observed difference between treatment and control group would be if the treatment had no effect. In non-experimental research, there is no treatment, thus the researcher must select a set of independent variables from non-experimental data that represent an unbiased data-generating model of the real world [25]. Only then is the *p*-value of a regression interpretable similarly to that of an experimental treatment. If there is anything misspecified about the model like confounding, suppression, or contamination, then the *p*-value is uninformative [26]. Researchers should also be aware that experimental research is also susceptible to confounding.

Moreover, if the data sample is not a simple random sample, then the reliability of the *p*-value decreases if not disappears. Most sociologists struggle to understand this. In a survey of communications researchers, 91% of students and 89% of postdocs and professors got most or all of the definition of a *p*-value wrong [27]. I am aware of no such survey in sociology, but it is reasonable to assume sociology is similar because communication research is highly sociological, often performed by those employed as sociologists. If you are not convinced, see how many of the Rinke and Schneider test questions you would have gotten correct, I myself would have done poorly prior to investigating this problem as part of my interest in open science. This is not meant to implicate myself or anyone; this is a serious problem in sociology that requires correction at the level of methods teaching. It also requires transparency so we can check each other's usages of the *p*-value to allow constructive criticism and increased awareness of its implications.

As a result of misinformation about *p*-values, researchers now regularly use the term "significance" and over time, a "significant coefficient" became synonymous with proof of a theory or significant knowledge about the social world. Again, a *p*-value indicates the likelihood that researchers would observe the same data if they repeated the same experiment several times, if the null hypothesis of no effect or no group difference was true. The *p*-value does not say anything about the effect size or meaningfulness nor anything about the theory, it only says something about the data collected; and only if the data-generating model is accurately represented by the statistical model. When *p* is less than 0.05 it means that there is a less than 5% likelihood of randomly observing the patterns found in the data, and therefore is evidence against the null hypothesis in the data. Whether the data and test reflect the real world cannot be determined at all by the p-value. Instead, sociologists routinely equate it with the true population parameter, and therefore assume that a *p* of 0.05 indicates a 95% likelihood of truth in their findings. A completely false conclusion.

By comparing *p*-values across sociological research we will discover the depth of this problem. This is an undertaking that has shed light on highly biased practices in many other disciplines. The need for this is obvious. If most social scientists do not understand

what a *p*-value indicates, yet nearly all quantitative social science is judged using *p*-values, there should be no surprise when social scientists misuse or abuse them and an entire field is biased and unreliable as a result.

Figure 1 demonstrates how *p*-values across many repeated studies should look under three different realities of a given population. The first reality would be no effect of a treatment or independent variable. If true, the distribution of *p*-values across all studies together on this population should look uniformly distributed like the grey dotted line. The second is a true effect. In such cases, there should be more and more findings with lower and lower *p*-values as shown by the skewed green line. If scholars report *all* of their studies on a particular topic in journals, or at least journals publish a random sample of these studies, then the distribution should follow one of the first two lines. If, instead, all published studies look more like the red dotted line, then there is some kind of human-induced bias. When studies' *p*-values hover just below the threshold, here shown as the break in density of results between 0.049 and 0.050, this is a likely indication of **p-hacking**—*the intentional efforts of scholars to submit only results with a* p-*value below a given threshold and to discard the rest*. Whereas a sudden drop off of values on the non-significant side of the threshold as shown with the large break in the red dotted line at 0.05 indicates **publication bias**—*the refusal of journal editors and peer reviewers to publish studies with non-significant results*. Publication bias only refers to the break in the red dotted line, not the curved distribution below the cutoff.

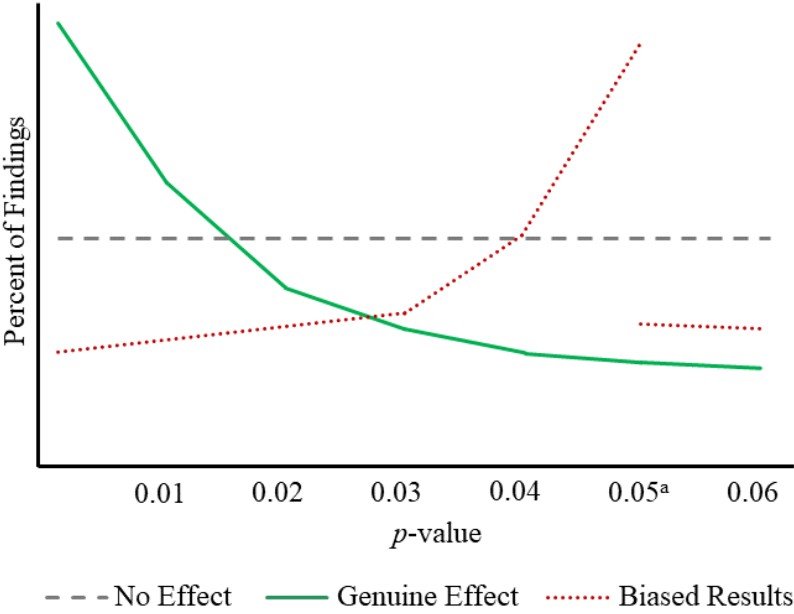

**Figure 1. How to visualize bias in *p*-values.** [a] This could be any pre-determined threshold such as *p* < 0.01 or 0.001 and result in the same expected patterns. Just that 0.05 happens to be one of the more popular 'cutoffs'.

Recent efforts at studying this phenomenon offer evidence of bias, where *p*-values across topics follow the red dotted line. For example, Head et al. [28] text mined *p*-values out of all open access papers in the *PubMed* database. As social sciences and especially sociology rely mostly on pay-walled publication systems for their top journals, these results are an irregular sample. Nonetheless, Head and colleagues find evidence of bias in the *p*-values. This p-hacking is present in several disciplines and highest in the cognitive sciences. There is no clear evidence of this in studies of "human society", but there were less than 200 cases leaving concerns about inference due to a small-N. Similar results supporting the biased red dotted line in Figure 1 were found in studies by Simonsohn, Nelson, and Simmons in psychology [29,30] and Brodeur, Cook, and Heyes in economics [31].

In sociology, an earlier study by Wilson, Smoke, and Martin [32] found that 80% of studies published in the top three sociology journals of that time rejected the null hypothesis, in other words they had *p*-values below a threshold. This suggests publication bias, if not p-hacking. Sahner [33], (in Table 5 in his publication) analyzed all article submissions to the *Zeitschrift für Soziologie*, 1972–1980. Of those that contained significance tests, 70% were significant at $p < 0.05$ suggesting that authors prefer to submit significant results. More recently, Gerber and Malhotra [34] reviewed articles published in *American Journal of Sociology*, *American Sociological Review* and *The Sociological Quarterly*, and specifically looked at the boundary of t = 1.96 (i.e., $p < 0.05$) to find that as many as 4-out-of-5 studies were 'significant'. This suggests publication bias as well. Sociology has yet to have a systematic review of p-hacking by comparing *p*-values within 'significant' results. However, there is good evidence that the perverse incentives to publish-or-perish among scholars and the perverse incentives to accept 'significant' findings among journal editors leads to bias toward results that appear attractive.

If playing with data to produce researchers' preferred results is not bad enough, researchers might only pursue a hypothesis test in a dataset *if* they observe some 'desirable' pattern in that data first. In doing this they have invalidated the p-test in the first place because they have conditioned their test on prior factors that do not represent the distribution of potential outcomes, i.e., they inflated the Type I error by some (most likely unknown) amount as discussed in Devezer et al. [35].

### 3. Sociology Needs Open Science, Just Like the Other Disciplines

The evidence presented in the previous section suggests that sociology suffers from problems common across other social science disciplines, problems that lead to unreliability. The next logical question is whether sociology, as a discipline, has a commitment to being reliable. A broad concept of sociology includes studies that are ethnographic, grounded, anthropological, ethnomethodological, self-as-case-study, and critical or oriented at power structures, and most of these are scientific in the sense that they are committed to rigorous method, logical argument, and observation. However, some are explicitly not 'science' or reject the entire notion of science and reliability. The authors of such work, therefore, have no interest in replicability or reliability, if they have data at all. This paper has no position on such sociologists, and I have no intention to define whether they are or are not sociologists. Their work is not for policymaking, knowledge transmission or for building a credible and transparent resource for others to follow, and that is ok and may have artistic and philosophical value. For the rest of sociology, I argue we need open science because (a) sociology is not unique and (b) sociology according to some of its most important voices should follow open science principles.

### 3.1. Sociology Is Not Unique

The largest sociological associations over the last century have been in Germany, the United States, and Japan. Next century will be India and China, but I focus here on these older sociological associations as core institutions of the global development of the sociology discipline. Each of the three has clear guidelines about sharing data, ethical practices, and reproducibility, similar to the other social science disciplines, see Table 1.

An estimated 78% of the major sociology journals have long-standing transparency policies that mirror those in Table 1 [36]. Unfortunately, these policies are mostly artifacts on paper without much enforcement. For example, only 37% of sociology articles published in the mainstream journals in 2012–2014 include shared data and/or materials [36]. In 2015, a small group of sociologists tried to obtain materials from the authors of 53 prominent sociological studies. They obtained these from just 19%, and only 20% of all the authors they contacted bothered to respond at all despite several requests [37]. These are unambiguously studies that used quantitative methods, but the problem appears to cut across methodological approaches. For example, a review of 55 qualitative interview-based studies from a top management research journal found that none of them shared sufficient

information to allow another scholar to engage in a "conceptual replication", as in repeat the study with the same methods [38]. This evidence suggests sociologists are free to hide the data and materials that led to their findings without recourse, despite guidelines.

**Table 1.** Transparency in sociological association ethics codes.

| |
|---|
| *American Sociological Association* 'Code of Ethics' 2019, 12.4.d.—Consistent with the spirit of full disclosure of methods and analyses, once findings are publicly disseminated, sociologists permit their open assessment and verification by other responsible researchers, with appropriate safeguards to protect the confidentiality of research participants. |
| *German Sociological Society* 'Ethik-Kodex' 2019 I.1.2—[When presenting or publishing sociological findings, they have to be described without omitting important results; i.e., that would falsify the findings. Details of the theories, methods and research designs that are important for the assessment of the research results and the limits of their validity are given to the best of one's knowledge.] |
| *Japanese Sociological Society*, 'Code of Ethics' 2019, Article 9—[Members must maintain open attitudes and behaviors to ensure a place for mutual criticism and verification.] |

[Author's translation]; for original language see Appendix A.

Many associations and publishers are addressing this problem through the adoption of the *Transparency and Openness Promotion Guidelines* (TOP) [39]. The TOP guidelines with help of the Center for Open Science support journals to improve science. Journals can become signatories of TOP, and in doing so they either adopt and enforce new transparency guidelines or certify that they already meet certain transparency standards. Most of the well-known psychology journals and several political science journals signed on. Other major journals such as the *Journal of Applied Econometrics* and later the *American Economic Review* adopted their own analogous and enforced transparency guidelines.

Until 2017, the only higher-ranking sociology journals that signed TOP were *Sociological Methods and Research* and *American Journal of Cultural Sociology*. In 2017, Elsevier (Amsterdam, The Netherlands) dictated that all its journals adopt guidelines, and this added *Social Science Research* to the list (more about Elsevier in Appendix E). At the time of writing this, the flagship journals *American Journal of Sociology* and *American Sociological Review* neither signed TOP nor enforce their own guidelines. Of top German sociology journals, the *Kölner Zeitschrift für Soziologie und Sozialpsychologie* is the only signatory, and no Japanese sociology journals signed.

If intransparency is pervasive in sociology, then research cannot be (a) checked for errors, (b) reproduced, or (c) simply critiqued. Even when exact reproducibility is not the goal, as often is the case with context-specific interpretive research, readers must take a giant leap to trust what others report when methods remain shrouded in mystery. Part of the problem is that sociologists express little interest in reproduction or checking others' works. There are few replications in the history of sociology, and if anything, they decreased over time until recently [40]. For example, searching the articles in the *American Journal of Sociology* and *American Sociological Review* reveals an average of 0.73 replications published per year (22 total) from 1950–1980, only 0.27 per year (8 total) from 1981–2010, and then 0.89 per year from 2011–2020 (see the ReplicationWiki for a list https://replication.uni-goettingen.de/wiki/index.php).

Looking at the popular usage of the terms *sociology* and *science* from a linguistic perspective suggests that sociology is a science. It is the task of philosophers and methodologists to argue at length about things like, 'what is science'; but dictionaries and the usage of language on a day-to-day basis offer an insight into sociology's public ontologies, and these are of the highest interest here. Major dictionaries' definitions in English, German, and Japanese appear in Tables 2 and 3 and suggest sociology is a science in popular denotation (Online dictionaries Merriam-Webster and Oxford (English), Duden (German), and Weblio (Japanese) accessed 30 August 2019.

**Table 2.** Sociology, a science by definition.

| |
|---|
| English—The science of society, social institutions, and social relationships; specifically: the systematic study of the development, structure, interaction, and collective behavior of organized groups of human beings. German—[Science; study of the coexistence of humans in a community or society, of the manifestations, developments and regularities of social life.] Japanese—[A science that seeks to clarify the mechanisms of social life, social organization, social problems, etc., in relation to human social behavior.] |

[Author's translation]; for original language see Appendix B.

**Table 3.** Science, practical and reliable by definition.

| |
|---|
| English—The intellectual and practical activity encompassing the systematic study of the structure and behavior of the physical and natural world through observation and experiment. German—[Reasoned/logical, orderly, reliable knowledge-producing research activity in a particular area.] Japanese—[A system or exploration of rational knowledge aiming at formally understanding areas of the world such as nature and society. It features facts based on experiments and observations, and systematic consistency based on logical reasoning. It is classified into natural science, social science, humanities, etc. according to the difference in research object and method. In the narrow sense, it refers to natural science.] |

[Author's translation]; for original language see Appendix C.

The "-ology" in the word sociology derives from Latin ("logos") to denote *the study of* something. Sociology is the study of the social, or *the science of the social* consistent with all three definitions in Table 2. Sociology is a science just like political science, psychology, social psychology, and economics. If sociology is considered science from an ontological and popular linguistic standpoint, the next question is what is science in the same popular definition approach. Looking at such definitions in Table 3 suggests *science* is a systematic, practical, logical, and reliable study of things in the public, common uses of the word across the English, German and Japanese languages.

In a 1983 speech, the economist George J. Stigler explained why disciplines such as sociology do not have a Nobel Prize. His reasoning was, "that they already had a Nobel Prize in literature" [41], p. 25. Such stereotypes reflect deep misunderstandings of what sociology and science are (see Tables 1 and 2), but at the same time indicate a poor reputation of sociology. Despite common stereotypes, economics is not more scientific than sociology. Economists use socio-economic data to test and develop theories about the world just like the other social science disciplines. In fact, their over-reliance on formal equations, like an over-reliance on p-values, may lead to unreliable results [42]. The problems are not entirely from outside sociology. Internally, some sociologists look down on others, and attempt to exclude them from sociology or sociological funding. This often can be reduced to some form of naturalism versus interpretivism, and resentments that emerged over recent trends offering greater funding for quantitative research. Any internal conflicts surrounding these inequities or divisions can only further damage the perception of sociology among other disciplines and the public.

*3.2. Open Science Is Sociology*

A movement is underway across science to correct the problems identified thus far. Various academics, associations, and funding agencies are now pushing to "open" science. To make it more transparent, reproducible, accessible, and less biased. The various efforts fueling this process can collectively be labeled the Open Science Movement (OS Movement) and suggests individual and institutionally sanctioned ethical practices, most often the sharing of all research materials, preregistering research plans, open access (to results and data), eliminating publication bias and 'hacking' to get results, and a renewed interest in replication. This movement arrived in the social sciences only in the last decade or so. In a short period, the OS Movement led to rapid changes in psychology, and

initiated incremental changes in other areas such as international relations, economics, communications, political science, and social psychology.

The basic ideas of open science are consistent with the ideals of three of sociology's key historical contributors (again see a brief review of what open science is in Appendix D). Robert K. Merton repeatedly argued for openness in science, including communal research for the public good. In fact, he is often cited in disciplines outside of sociology because he studied science so extensively; but he is an obvious choice to discuss here. Max Weber and Jürgen Habermas have linkages to open science ideals that are perhaps less well known or apparent on the surface, and whose different takes on social science between a more positivist versus critical and interpretivist epistemology are striking. I review some of these three scholars' ideas here in an attempt to locate open science as a part of sociology based on critical self-definitions.

Merton's ideas provide a foundation for open science as sociology. According to his idea of *organized skepticism* sociologists must consistently certify the knowledge they produce. He argued this should follow communalism, where they must do *communalism* they must do this in a community of sociologists (Merton used "communism" in the original text, but this word has a different connotation today that may cause confusion). This definition demands that all scientists have access to the same knowledge or materials of knowledge construction and have the opportunity to participate in scientific exchanges without these exchanges taking place in secret, following his concept of *universalism*. All together, these norms call for open access, open data and transparency (communalism and universalism), and reproducibility (communalism and organized skepticism) identical with most definitions of open science and the OS Movement. Merton proposed these norms as a paradigm shift during his time, and the OS Movement proposes similar norms today calling for a similar paradigm shift (not only in sociology but across all science, see for example Chubin [43]).

Some decades before Merton, Weber stated that the task of a sociologist is to provide facts while engaging in critical self-clarification [44], p. 505. He was careful to define "facts" as both objective and subjective—also evidence that Weber was not purely a positivist as many might believe. Weber argued that the effective sociological teacher avoids imposing subjective ideology (e.g., political or cultural preferences) when presenting facts to students or colleagues (*Werturteilsfreiheit*: 'value-free'/'judgement-free'). Each sociologist needs to have in mind their own subjective interests and goals and how these relate to the facts they present in order to do this. Weber proposed therefore that sociologists cannot separate the object of inquiry from the method of inquiry, namely the sociologists themselves cannot be entirely exogenous to the social world they observe (*verstehendes Erklären*; something like 'self-awareness' and 'clarification' in the research process [45], p. 495). To maintain this self-awareness as a researcher in the process of research, a community of researchers is necessary to provide insights that lead to self-awareness through feedback. The impact of a community is limited to transparent and continual critique (as Merton also argued). For example, the hoax papers by Lindsay, Boghossian, and Pluckrose were so attractive subjectively, that editors and reviewers rarely bothered to question their methods or data.

Perhaps surprisingly to some, I believe that Habermas makes claims that are consistent with the OS Movement. He continuously insisted on a normative change at the level of communication. He advocated for a rationalization of communication that would open up all communication for consumption. This would reign in individuals' intentions to perform communication to serve their own interests and promote instead a goal of understanding between individuals [48], pp. 94–101. When the public sphere is commodified and controlled by private interests, there is asymmetric production of communication (those who control the communication channels control the content) and asymmetric consumption (higher quality communications become more expensive) [49]. This distorts communicative action. If I switch the word "communication" with "science" or "sociology" then open and transparent communication would reshape science in the same ideal way that Habermas imagined for society. After all, science is part of society, so the OS Movement could be seen

as simply a sub-section of a larger Habermasian movement in all social spheres. When the common values of science shift toward more openness it can solve the 'pathologies' of science, namely profiting from closed knowledge communications and the rigid norm of publish-or-perish.

These three sociologists are monumentally influential, but worlds apart in their philosophies, arguments, and research; Habermas was far more than a sociologist and is cited across many disciplines, not only sociology. Merton was a positivist, and Weber was mostly a positivist, describing the world as he encountered it; whereas Habermas was largely an interpretivist and described a world with hidden meaning and power structures. However, they three similarly expressed concern about scientific misconduct and the influence of private interests. A simple reading of their basic claims suggests a more responsible science if not responsible social action on the whole. Weber, like Merton, asserted that science as a vocation can be problematic. Individuals may place vocational goals ahead of scientific goals, and this can impact the reliability of their results. Habermas, like Merton, argued that morality and social utility are necessary scientific research norms.

All three theorists set forth ideal sociological practices that are still unrealized by the discipline today. Each in their own way argues that without meeting these ideals, sociological knowledge is not useful. In my view, production of useful knowledge is a necessary condition for practicing sociology and consuming public funds in this process. Of course, sociology will never be perfect, but if we choose not to decide to pursue open science, we still have made a choice—an inaction whose implications may be harmful in ways we cannot imagine further into the future.

Weber called on scientists and teachers to expose inconvenient facts without promoting one or another position. In this essay, I tried to present what I believe to be an objectively inconvenient fact in sociology: Our rich universe of theory and critique is unsubstantiated, because we only rarely provide the materials for, and engage in, this substantiation process. Sociology is not special in this regard; however, other neighbor disciplines have taken the lead in reforms. This risks putting sociology at a (further) disadvantage in terms of public trust, funding, and contributions to science as a whole. Therefore, I see my 'position' in favor of open science as a practical response to the *status quo* in sociology.

To me this position is obvious given the reality of sociology. When papers are not scrutinized then they are not reliable. They do not consist of findings but illusion and possibly sophistry. This is not a useful knowledge. We cannot discern whether this is just but a bunch of academics playing a game to win notoriety and citations, push particular private agendas, or actually science.

The prestige one can gain through academic work is theoretically unlimited. This potential for prestige accumulation produces "ego-maniacs" [13], p. 1358. Bruno Frey calls the academic institution a system of "Publishing as Prostitution", because we face a monumental tradeoff between pursuing our "Own Ideas and Academic Success" [46]. Perhaps in an obfuscated act of irony that motivates open science (or simply evidence of how easy it is to be unethical without really trying), Frey got into trouble after he published a similar paper in four different journals [47]. If we as sociologists, or any scientists for that matter, cannot provide useful knowledge ('ideas') to the social world, then taking the OS Movement seriously would lead to a call on public and private funding agencies to discontinue supporting us sociological researchers. This is not a radical or self-defeating position, unless our only reason for practicing sociology is to have a comfortable job with research funding. By supporting this position, just like being transparent before and after we conduct our research, we ensure our goals are ethical and target useful information as opposed to private status accumulation, in line with the sociology Weber and Merton argued for.

## 4. Resistance to Open Science

Achieving open science ideals of the OS Movement and laid out by Weber, Merton, and Habermas will not be easy in sociology. Researchers using qualitative methods are

more often skeptical of open science. Given that qualitative methods often involve meaning-making between the researcher and subjects, and a sensitive and contractual nature of data collection, a degree of reluctance is understandable. Two studies, one in Germany in 2004 [50] and another in Switzerland in 2012 [51], offer insights into this reluctance (as reported in Herb [52]). In both countries, around half of the respondents (60 and 47%, respectively) were willing to share data but only 15% would do so publicly, while the remainder would have conditions or restrictions placed on the data. However, respondents tended to believe that the written or verbal consent attained from their participants legally prevented them from sharing the data.

These surveys also revealed that current practices in qualitative research do not include sharing data as part of the research design. Although about half of the surveyed researchers were willing to share data, less than 20% ever asked for their participants' consent to do so. When researchers do not plan to share data by default, the problem is procedural leading many researchers who could share their data to simply not do so out of custom or training. Of course, sharing information that would jeopardize the well-being of participants is unethical and against the positions of sociology associations and universities. The OS Movement, to my knowledge, has no intention to undermine the ability of qualitative researchers to access certain populations that require building trust and guaranteeing anonymity. Unethical behavior is certainly inconsistent with Mertonian ideals of science. But, there are certainly a host of studies that could have been planned with sharing in mind, and this constitutes a lost opportunity to promote both future research and reliability.

Researchers often question why data sharing is so important. The answer relates to bias and hacking. Researchers using qualitative methods cannot simply p-hack, but they can take phrases and words out of context and construct or interpret things to their liking. A study by Riemann (2003) published in *Forum: Qualitative Social Research* ("Qualitative Sozialforschung") asked several researchers to qualitatively analyze the same narrative autobiography data. The results went in different directions with little in common (see: http://www.qualitative-research.net/index.php/fqs/issue/view/17).

The Riemann study is perhaps not surprising, and simply an indicator of a process where different insights regarding the same social worlds are precisely what moves science forward. New ways of seeing, knowing, framing, and constructing are the backbone of the historical progression of sociological knowledge. As meaning-making between researcher and subject(s) constitutes a unique moment, it is impossible for another researcher to have the same experience and construct the same meaning. Here open science is not a one-size-fits-all. Transparency and replicability diverge between qualitative and quantitative methods. For studies with qualitative methods, enough information should be available that another researcher can understand the logic and process behind the meaning-making. Having a plan that is announced in advance is extremely helpful, just like a researcher pre-registering a planned experiment or survey. Without a plan and a theoretical perspective, results of qualitative research appear selective and unreliable, even though they may not be. Without transparency it is impossible to know if researchers had a plan, what they expected to find, and exactly how their specific methods would answer a particular research question. Without this information it is impossible to know the true meaning of their results, regardless of method. In the Riemann study, we can only say today that we learn about how different researchers look at identical narrative data, we cannot, and in fact should not, say much about the person who gave the narrative. Without a study design and context, as in what was happening when the narrative was given and why was the narrative given, we might as well be analyzing fiction.

Transparency for qualitative methods has different considerations than quantitative methods. Each unique method may entail unique transparency challenges. Nonetheless, eminent proponents of qualitative methods concluded in the *Workshop on Interdisciplinary Standards for Systematic Qualitative Research* organized by Michèle Lamont that transparency is essential. The first two recommendations of this workshop were:

- "Situate the research in appropriate literature; that is, the study should build upon existing knowledge", and
- "Clearly articulate the connection between theory and data" [53], p. 4.

Here "existing knowledge" refers to methods, logic, and previous studies. When these things are considered prior to conducting the research and made transparent before and after conducting the research, open science ideals of transparency are resoundingly fulfilled regardless of whether the participant's identities, locations, or specific statements are shared. Qualitative researchers often work in teams to collect data and this already requires careful coordination and transparency among researchers, and many qualitative researchers share their materials under careful supervision, analogous to secure data centers that attach quantitative survey data to neighborhoods or genetic data and require a security clearance to access. Moreover, successful qualitative researchers depend on their own meticulous archiving of methods and data because they will re-examine the data and publish many findings from these same data. Therefore, in some areas that are heavily dependent on qualitative methods, open science ideals are already in place [54]. That all information regarding the methods and goals of researchers are necessary for understanding and trusting the relevance and meaning of their findings is nothing unique to the OS Movement. It is simply following the scientific method and the methods of sociology.

Doing this carries potential benefits across science and makes further scientific discourse possible. For example, Cramer [55] studies public opinion using qualitative methods, and in her studies, she is ethically prevented from sharing her particular participants' data. However, her work is a model of how to do open science with qualitative research because she makes everything but the data transparent. Here transparency means describing the process in detail. Moreover, because public opinion scholars predominantly use quantitative methods, she has to be extremely transparent and carefully describe her methods in order for them to understand and see that her findings are useful among a quantitatively dominated field. Researchers using qualitative and quantitative methods alike cannot possibly share their entire workflow and fit within the word limits of journal articles. This means online sharing is necessary; by now, the resources for this are easy to find and nearly unlimited.

A taskforce of political scientists investigated whether researchers using qualitative methods should support transparency guidelines [56] pp. 20–21. They argued that transparency policies would impose a positivist understanding of science on researchers and narrow the realm of what constitutes "research". This suggests that, so far, the OS Movement failed to offer clear ideas and solutions about transparency for those working in qualitative research. A huge red flag in the idea of transparency guidelines is offering any particular group of researchers the opportunity to diminish the value or reputation of other groups as opposed to giving them an opportunity to stand on equal empirical and logical footing. For example, if any data guidelines create policies that render qualitative methods 'unethical' or 'out of line' in some way, as if to frame research using qualitative methods as somehow abnormal, then science is not open. If it is the goal of any groups claiming to represent the OS Movement to disavow certain methods or fields, then these groups need the ethical tenets of open science as much or more than anyone else.

Transparency in open science is a way for *all* researchers to reveal the process, ideas and materials on which an argument or theory is founded, and to support a more ethical scientific community, even if this is only possible prior to the conducting of a study. In fact, the TOP guideline, "5 Design Analysis Transparency" promotes making a statement on data sharing; in its ideal this means authors must state what they share and why. Again, with ethical reasons, qualitative researchers can withhold sensitive data, and simply make a statement why, while also making a statement about why they do or do not share all steps in their design and research process. This is not a high, or exclusive bar to pass to get published in a journal. Moreover, if the design is presented to colleagues in advance for discussion, like in a colloquium, and submitted to an institutional review board, then the

study becomes more of a communal effort and potential ethical concerns are not left solely to the individual researcher.

It seems that a practical definition of open science should follow what George Soros and earlier Karl Popper and Henri Bergson argued for as *open society*; namely, a society capable of continual improvement. One with public transparency of government and business activities, one where knowledge is accessible and entrusted to all, and one where groups are not excluded from fully taking part in the society. If applied to sociology, it would lead to Mertonian scientific norms and Habermasian open communication. A science, and in particular a sociology, capable of continual improvement through a commitment to transparency and institutions designed to promote this transparency at all costs, is a science that the public should want to fund.

## 5. Conclusions

An ultimate goal of sociology is the production and refinement of theory to predict, explain, demystify, challenge, or (re-)interpret features of the social world. Theories are the building blocks of social change. If we do not have useful theories of poverty, discrimination, group identity, or climate change for instance, then we cannot tackle related social problems. The difference between a theory and an assumption is reliable research. Given current sociological practices we cannot be sure the knowledge we produce is reliable, for example when the research process and/or resulting data are hidden, then any mistakes or scientific misconduct are unknowable. Theories derived from unknowable, unreliable results are unreliable. Practicing transparency allows you, me, us, and them to produce reliable theory and, when we all do it, sociology is reliable.

In practice, achieving this reliability is a major challenge. Of course, if all researchers engaged in ethical, open science practices the problem would be nearly eliminated. Unfortunately, without institutional and structural changes, scholars practicing open science will enable some individuals to deceive and cheat sociology. In the absence of ethical or quality standards, the highly competitive nature of science makes such actions rationally self-interested at least for the nefarious and sociopathic minded. Such individuals could ride on the coattails of a new and improved public image of social science created by a grassroots open science movement. Therefore, the incentive structures must change. Enforcement of transparency requirements (with explicit protection against the identities of vulnerable populations); public announcement of studies in advance with clear expectations, target populations and research designs; and an immediate end to for profit publishing would move sociology closer to an ideal advocated by Weber, Merton, and Habermas. This would also lead sociology to easily meet its own associations' guidelines for ethical and best practices. However, the question of what is ethical and who gets to decide is not an easy one to pose, and inevitably leads to divisions and power struggles. Therefore, I hope this paper provides a first step toward greater awareness that can foster new cohorts of scientists prepared in advance to do science that is more reliable and thus practically and perceptibly more useful to policymakers and society at large.

**Funding:** This paper was partly developed as a Wikimedia Freies Wissen Fellow, Cohort 2019-20.

**Institutional Review Board Statement:** Not Applicable.

**Informed Consent Statement:** Not Applicable.

**Data Availability Statement:** Not Applicable.

**Acknowledgments:** In addition to support from the Freies Wissen Fellows community, I also thank the Berkeley Initiative for Transparency in the Social Sciences for allowing me to act as a Catalyst. A German version of this paper is available, "Braucht die Soziologie Open Science?". An earlier version of this paper was given as a keynote speech at the 7th Student Sociology Congress at the Ruhr-University in Bochum, Germany (Studentischer Soziologie Kongress) on 22 September 2019. I am thankful to discussions at the Open Social Science Conference (OSSC19) at the University of

Mannheim 25–27 January 2019 and helpful comments from Jan Höffler and Judith Offerhaus, and insightful and humbling comments from a blind peer reviewer.

**Conflicts of Interest:** I am on the *Societies* Editorial Board. In terms of science, I do not see this as a conflict of interest because open science is my main motivation for writing this paper and the reason that I decided to serve on the board to promote change from within this emerging, open access but for profit, journal. I receive no compensation for this editorship. This paper is considered a "Communication" rather than an empirical or theoretical article, although it did undergo peer review.

## Appendix A

**Table A1.** Original Language.

---

*Deutsche Gesellschaft für Soziologie* ‚Ethik-Kodex' 2019 I.1.2—Bei der Präsentation oder Publikation soziologischer Erkenntnisse werden die Resultate ohne verfälschende Auslassung von wichtigen Ergebnissen dargestellt. Einzelheiten der Theorien, Methoden und Forschungsdesigns, die für die Einschätzung der Forschungsergebnisse und der Grenzen ihrer Gültigkeit wichtig sind, werden nach bestem Wissen mitgeteilt.
(https://soziologie.de/fileadmin/user_upload/dokumente/Ethik-Kodex_2017-06-10.pdf, accessed 13 September 2019.)
[When presenting or publishing sociological findings, the results are presented without falsifying omissions of important results. Details of the theories, methods and research designs that are important for the assessment of the research results and the limits of their validity are given to the best of our knowledge.]
*American Sociological Association* ‚Code of Ethics' 2019, 12.4.d.—Consistent with the spirit of full disclosure of methods and analyses, once findings are publicly disseminated, sociologists permit their open assessment and verification by other responsible researchers, with appropriate safeguards to protect the confidentiality of research participants.
(https://www.asanet.org/code-ethics, accessed 13 September 2019.)
[Entsprechend dem Geist der vollständigen Offenlegung von Methoden und Analysen gestatten Soziologen nach der Veröffentlichung der Ergebnisse ihre offene Bewertung und Überprüfung durch andere verantwortliche Forscher, wobei angemessene Schutzmaßnahmen zum Schutz der Vertraulichkeit der Forschungsteilnehmer getroffen werden.]
Japanese Sociological Society, Code of Ethics' 2019, Article 9—< 相互批判・相互検証の場の確保 > 会員は、開かれた態度を保持し、相互批判・相互検証の場の確保に努めなければならない。 (https://jss-sociology.org/about/ethicalcodes/, accessed 13 September 2019.)
[Platz für gegenseitige Kritik und Überprüfung sichern] Die Mitglieder müssen eine offene Haltung bewahren und sich bemühen, einen Platz für gegenseitige Kritik und Überprüfung zu gewährleisten (die Verifizierung).
[(Securing a place for mutual criticism and verification) Members must maintain an open attitude and behavior to ensure a place for mutual criticism and verification.]

---

## Appendix B

**Table A2.** Original Language.

German—Wissenschaft, Lehre vom Zusammenleben der Menschen in einer Gemeinschaft oder Gesellschaft, von den Erscheinungsformen, Entwicklungen und Gesetzmäßigkeiten gesellschaftlichen Lebens.
[Science, teaching of the coexistence of people in a community or society, of the manifestations [institutions?], developments and laws of social life.]
English—die Wissenschaft der Gesellschaft, der sozialen Institutionen und der sozialen Beziehungen; konkret: die systematische Untersuchung der Entwicklung, Struktur, Interaktion und des kollektiven Verhaltens organisierter Gruppen von Menschen.
[the science of society, social institutions, and social relationships; specifically: the systematic study of the development, structure, interaction, and collective behavior of organized groups of human beings.
Japanese— 人間の社会的行為と関連づけながら、社会生活・社会組織・社会問題などのしくみを明らかにしようとする学問。
[Eine Wissenschaft, die versucht, die Mechanismen des sozialen Lebens, der sozialen Organisation, der sozialen Probleme usw. in Bezug auf das soziale Verhalten des Menschen zu klären]
[A science that seeks to clarify the mechanisms of social life, social organization, social problems, etc., in relation to human social behavior.]
[Author's own translations], *NOTE:* 学問 can also be translated as 'area of study'.

## Appendix C

**Table A3.** Original Language.

German—(ein begründetes, geordnetes, für gesichert erachtetes) Wissen hervorbringende forschende Tätigkeit in einem bestimmten Bereich.
[Logical, orderly, reliable knowledge-producing research activity in a particular area.]
English—the intellectual and practical activity encompassing the systematic study of the structure and behavior of the physical and natural world through observation and experiment. (This was taken from Oxford dictionary because Merriam-Webster offers a first definition of *science* as, "the state of knowing: knowledge as distinguished from ignorance or misunderstanding" and only in the third definition does it address the practice of science, as knowledge, " . . . especially as obtained and tested through scientific method". But to get the 'science-in-practice' definition one also needs to look up "systematic" and "scientific method", whereas Oxford's definition has these concepts included without further need to look up words.)
[Die intellektuelle und praktische Tätigkeit umfasst das systematische Studium der Struktur und des Verhaltens der physischen und natürlichen Welt durch Beobachtung und Experiment.]
Japanese—自然や社会など世界の特定領域に関する法則的認識を目指す合理的知識の体系または探究の営み。実験や観察に基づく経験的実証性と論理的推論に基づく体系的整合性をその特徴とする。研究の対象と方法の違いに応じて自然科学・社会科学・人文科学などに分類される。狭義には自然科学を指す。
[Ein System oder eine Erforschung rationalen Wissens, das darauf abzielt, Bereiche der Welt wie Natur und Gesellschaft formal zu verstehen. Es enthält Fakten, die auf Experimenten und Beobachtungen beruhen, und systematische Konsistenz, die auf logischen Überlegungen beruht. Es wird in Naturwissenschaften, Sozialwissenschaften, Geisteswissenschaften usw nach dem Unterschied in Forschungsgegenstand und Methode. Im engeren Sinne bezieht es sich auf die Naturwissenschaft.]
[A system or exploration of rational knowledge aiming at formally understanding areas of the world such as nature and society. It features facts based on experiments and observations and systematic consistency based on logical reasoning. It is classified into natural science, social science, humanities, etc. according to the difference in research object and method. In the narrow sense, it refers to natural science.]
[Author's own translations]

### Appendix D. What Is Open Science?

Although it has no global definition, advocates of "open science" tend to agree on some points. Most definitions of open science floating around the Webs, focus on the practical aspects of accessibility. For example,

" ... the practice of science in such a way that others can collaborate and contribute, where research data, lab notes and other research processes are freely available, under terms that enable reuse, redistribution and reproduction of the research and its underlying data and methods."

(FORSTER, open science teaching resource)

Some definitions enter the realm of ethics, feminism, and social justice. For example,

" ... to imagine and design inclusive infrastructures, practices, and workflows for scientific practice that intentionally enable meaningful participation and redress (these new) forms of exclusion."

(Denisse Albornoz, OCSDNet)

Whatever the ontology, open science is inevitably something that challenges the status quo in science. Usage of term indicates there is something undesirable about science. Some refer to this as a 'reproducibility' or 'reliability' crisis (a detailed and useful report was recently put out by the EU on this [57]. If there was nothing wrong with science, then we would simply advocate "science". The "open" part of the concept refers to any number of things depending on whom you ask. Commonly it means:

**Open access**—making the results of scientific techniques, research, and theory accessible to everyone, as opposed to only in paywalled journals.

**Transparency** <open process>—making all methods, code, data, and any biases or conflicts of interest known before and after the research is conducted. So long as doing this does not harm human subjects or violate any laws.

**Open source**—on the technology side of science, all programs, apps, algorithms, tools, and scripts should be transparent and usable by others. This means that when a scientist develops a new technology, anyone else's technologies can interact and interface with it. Moreover, anyone can modify the technology to better suit their own needs.

**Open academia** <open communication/democracy/feminism>—allowing anyone to participate in academia. That inequalities, prejudice, and domination that take place in the social world are eliminated from the academic world, if not that the academic world has a goal of eliminating them in the social world. That everyone has the same place in scientific discussions, and no science is conducted by pressuring others or taking advantage of existing power structures. That no science takes place in secret, except for research that requires obfuscation for its completion.

Again, the definitions can cover a broad range. The above are just a snippet, although they strike me as the most common usages; except for 'open academia', this is reserved for certain justice motivated scholars, but it is important to recognize that any form of "open science" that reinforces academic privilege already existing in the English-speaking and Global North countries is not "open" [58].

### Appendix D.1. Double-Work and the Co-Opting of Journals

Scientists provide their work as editors and reviewers, because the peer review and publication process is the centerpiece of all of science. Peer reviewers and editors are the only consistent form of quality control in science. The academic journal was a functional response to previous forms of knowledge transmission that required direct scientist/practitioner to student interactions, which were geographically limited and reached a very narrow audience.

The journal made it possible to transmit knowledge across the globe. Moreover, the journal reduced the simultaneous discovery and re-discovery problems of science, because no one could prove they discovered something first, and others worked on problems that were already solved unknowingly. It represents one of the first 'open science' movements

because it was driven by the idea that science was at an impasse and could only move forward through transparent and open exchange of ideas arbitrated by being part of the public record through publishing.

Ironically, the journal format came full circle and began to undermine science. After over two centuries of journals run by non-profit academic associations, for-profit publishing houses began 'offering' their services to meet the growing global demand for journals and their content and the rising costs of editing and distribution. In many cases, these publishing houses were able to purchase the journals by offering the academic societies the exclusive right to determine what went in them. Within just 30 years, five conglomerates owned the titles, content, or certain features of over 50% of all journal articles published globally.

The content, as always, is still a product of the scientists and the voluntary work of editors and peer reviewers. The publishing houses make large profits but pay nothing to these workers. The editors and peer reviewers earn their income from universities mostly. The very universities that pay high fees to purchase the right to provide the journals in their libraries. This is a double tax on the universities—paying the producers of content to produce and then paying the distributors of that content to consume it. The content does not change at any point in between these two forms of payment, in other words, the publishers do not add any scientific value to this content.

Matters got even worse with the publishing houses over the past decades. As creative and deceitful capitalists, the publishing houses realized they could generate even more profit by collaborating with the private sector. For example, pharmaceutical companies' profits were directly determined by the findings of studies published in journals. Pharmaceutical companies, or any companies whose profits were determined by the outcomes of scientific experiments, would be willing to invest in shaping those outcomes if they could. Enter a novel concept pioneered by Elsevier: Selling journals or journal space to private companies to boost their profits. Win–win for them.

*Appendix D.2. Publish-or-Perish Begets Questionable Research Practices*

Thanks to the advent of the scientific journal, knowledge could be evaluated, used, and further transmitted across space and time. The utility of the journal and other forms of academic publication such as books, proved so effective that they became the primary source for others to evaluate the importance of scientists and their work. This gave rise to the norm we are all familiar with, publish-or-perish.

In a survey of psychologists, John et al. [24] found that 50% claimed they had selectively reported studies that supported their hypothesis (as in, selectively excluding those that did not). Moreover, 35% admitted to reporting unexpected findings as having been predicted from the start. Nearly 2% outright admitted to faking data.

Publish-or-perish and questionable research practices have a causal relationship. Except for occasional sociopathic or psychotic individuals, there is no reason for a scientist to engage in questionable research practices. No reason, except scientists' very existence on scientists may depend on it. So, many studies in reality lead to results that go in all directions, support the null, or (most importantly) do not provide groundbreaking new results.

Through the peer review and editorial process, journals select studies that are pathbreaking; studies that will move knowledge forward and be of the greatest interest to readers. When faced with prospects of not getting tenured, not getting grant funding, and being forced out of academia, a human's (scientist's) rational calculations change. Suddenly, rounding that $p$-value from 0.054 to <0.05 or even adding some cases to the data becomes a cognitively defensible decision.

Like any profession, science is competitive. Those who publish more or get more citations to their publications tend to get ahead. Those who do not, do not. Professional athletes use incredible tactics to gain competitive advantage. Of course, steroids are well-known, but other tactics are much harder to detect. For example, endurance athletes often

use blood transfusions to boost recovery and performance. This is what it means to be human, scientist or not.

I suspect that most questionable research practices are not intentional. This pressure to find a job after doing doctoral studies and then to get tenured, means a tradeoff between conducting science in its ideal form—so learning as much as possible about the existing literature on a subject, mastering the necessary methods to perform the research and executing the research, possibly with several iterations, and facing the prospect of null results—with science in a form that will lead to publication as fast as possible.

This 'fast as possible' leads to amateur science. For example, in the rush to get my first publication, I attempted to use "multiple imputation" but lacked the time to properly learn this method. Instead, I simply generated several datasets and averaged them into one and re-ran the analysis on this one. This was not an intentional misuse of a method. It is a questionable research practice as a result of context. Think about matrix algebra. It is the basis of many advanced statistical techniques regularly used by social scientists. How many of us have a strong grasp of matrix mathematics? I do not; and yet I have published several studies using structural equation modeling.

*Appendix D.3. An Open Sociology Movement?*

Open science and the Open Science (OS) Movement embody the ideals of Weber, Merton, and Habermas as argued in the main text of this paper. The movement follows Merton and Habermas' calls for open and communal discourse. Therefore, if sociology is going to practice open science, and that this will arrive as a result of concerted action of the OS Movement, it means there shall be no exclusions. It means that open science in sociology should be something to unite sociologists because it asks them to make themselves and their discipline reliable and trustworthy. Although there are many takes on the OS Movement, if it is to be labeled "open" and is to follow the assertions of some of its most influential thinkers, then it must lay aside claims to philosophy of knowledge. It must not be a positivist movement. It is a movement to open the black box surrounding what sociologists do, whatever it may be, to create a community of quality control and dialog. It is also a movement to smash the paywalls blocking others from accessing findings, especially useful to integrating researchers in the Global South. It is a movement to increase ethical practices. If effective, its outcomes will extend beyond reliability and access. Open science will increase trust in sociology not only among the public, but also policymakers, other disciplines, and sociologists themselves as suggested by a recent survey tracing the impact of the OS Movement [59].

Many sociologists are unaware of the OS Movement, or they believe it is not relevant for them. Some think that because sociologists rarely conduct experimental research or often use qualitative and interpretivist methodologies, transparency and reproducibility do not apply to their research. Some think the OS Movement is a cult, a band of thugs trying to gain notoriety by disproving or shaming others [60]. Others might argue that open science is not important for sociologists because they do not cure diseases or plan space travel, for example. I believe these assumptions are based on misconceptions of the ethical and practical goals of the OS Movement if not an underestimation of the importance of sociology. In my view, open science is a paradigm shift that sociology must follow or accept its own failure as a scientific discipline. Sociology can effectively study and solve social problems, the kind that are far more insidious than the spread of a disease or development of a technology. For example, the study of group dynamics and conflict can reduce the spread of war or racism in theory, and the study of social inequality can increase social justice. Open science, at least in my opinion, is an avenue for sociology to gain legitimacy, and claim its place as a reliable science for solving social problems.

**Appendix E. Actions to Take Now**

Experiences from other disciplines suggest that open science practices are worth it. They are increasingly rewarded in academia, with growing funding opportunities, greater

exposure of results leading to career advancement, greater exposure of research and data leading to improved future science and helping other scientists, being a part of a movement to improve science has intrinsic/ethical benefits, and doing science in a more open and communal fashion can be enjoyable [61–64]. The nefarious publish-or-perish norm is probably the greatest threat to the possibility of open science, because science itself, in particular universities and governments, has not found a clear alternative and sustainable method for evaluating and promoting its scholars. The mental health of the scholars themselves as a result of this norm is at stake as much as the credibility of science [65], and open science should be a way to address both. Of course, this cannot come only from practicing sociologists and requires sweeping institutional changes [66], but there are many things we can do to affect these changes.

Here I speak directly to the sociologist (or any social scientist) researcher, about what you can do now to improve both your own work and sociology as a whole.

*Appendix E.1. Transparency*

Make all the materials associated with a research paper or book available online. This means research design, methodological steps, data (when legally and ethically possible), analyses, conflict of interest, and any software code. The practical reason is that others can follow your work and expand it in the future. Yes, this means that others may replicate or critique your work, but from a career perspective, you want others interested in your work regardless of whether they want to criticize or applaud it. In this process, you should support principles of constructive criticism, and whether the criticism you receive is useful to you depends on how well you react to it; for example, I replicated a study [6] calling its results into question, then Weakliem [67] replicated my study and called my results into question (https://www.sociologicalscience.com/articles-v3-6-109/). Thanks to an innovative open comment platform at *Sociological Science*, we were able to engage in constructive discussion thereafter. Constructive exchange can lead to collaboration with critics to generate better future research without personal conflicts. Being transparent forces you to be careful. Knowing everything will be public information increases the value of attention to detail. Put in its converse: Not sharing your workflow publicly can indirectly foster lower quality standards, in addition to creating possibilities for misconduct. All this enables rather than hinders knowledge and increases inter-researcher trust.

Transparency should not be much extra work. During the research process you should take high quality notes for yourself. You will often return to your data and research in the future and thus need those notes. This is a best practice with or without sharing your work. When you engage in this best practice, you have a deep familiarity with your data and can draw meaningful conclusions and easily redact identifying characteristics in your data. In case you cannot share data, you can still reveal the design and expectations; or allow controlled access to the data. Alternatively, you might reconsider whether you really need to anonymize. When you ask participants for informed consent, in some cases, you can ask permission to share details about their lives publicly and avoid the problems of anonymization in case the study does not involve risks to you or them [68].

The 'transparency work' of the qualitative research process can be reduced by software platforms that provide semi-automated annotation and coding [69]. Even if you do not share data, you can build an open workflow from the beginning that allows others to understand every step of the data generating process [70]. However, this work can also be extremely tedious and the incentives not immediately clear. More fruitful discussion if not research assistant funding is needed in this area moving forward.

If you are using quantitative methods, immediately stop hiding your work. If you ran 100 models and 99 did not support your hypothesis, then this is your finding. If a journal does not want to publish this, point the editors and reviewers to the importance of null results and the problems of publication bias. If they still refuse, consider boycotting this journal and sharing your negative experience in public.

*Appendix E.2. Preregistration*

Preregistration can drastically reduce bias and hacking prior to collecting data. When you clearly outline your plans including how you will analyze the data, before conducting the research, there is little room for hacking so long as you stick to the plan. Moreover, preregistration can be done directly with a journal. This means you write the article prior to data collection and then simply add the results later—a process known as *registered reports*. The journal has agreed to publish the article regardless of the results and therefore the incentive to hack is dramatically reduced. This agreement comes from a peer review process without results. This means peer reviewers cannot reject results that they do not like or that do not support their own research. In a preregistration you must think much harder about factors such as meaning, causality, inter-subjectivity, and 'how the world probably works'. You cannot hide behind results in this process and therefore you must anticipate counterarguments and explore counterfactual logic. This improves the clarity of theory and research, creating an immense gain in efficiency and effectiveness.

Regardless of the methods you use there are many opportunities to take advantage of preregistration. Some forms of qualitative research, for example those involving grounded theory and interpretivist methods, require decisions during the research process that cannot be foreseen. This uncertainty can be outlined in a preregistration stating explicitly when flexibility is and is not admissible [71]. Moreover, simply putting a qualitative research plan online prior to conducting the research is equivalent to a pre-analysis plan. This research design need not compromise your data collection work because you can register the plan on a platform like the Open Science Framework and then embargo it, so that it is preserved but not made public until after the research concludes. Some scholars using quantitative methods might assume that preregistration is not possible because they work with secondary survey data. However, the regularity and release of these survey data are known in advance, and these scholars can preregister their studies before the next round of data are collected with the knowledge of which questions and countries will be available.

*Appendix E.3. Decommodify Science*

The central functions of the scientific publishing industry are printing and disseminating knowledge, which historically solved a problem of how to share knowledge across universities and countries. The business functions of publishing, however, come with harmful byproducts. Publishing firms extract profits from scientists twice. First, scientists provide free labor in the form of editing and peer reviewing, in addition to producing the results for the articles to be printed. Next, researchers, or their employers, must purchase the product of their own labor; labor not paid for by the publishers. The journal article as a product comes at a high cost, and often only in packages of journals meaning that universities have to pay for extra material their scholars do not use [72].

Sometimes publishing houses neglect science in favor of profits, but Elsevier has been particularly problematic. They sponsored weapon fairs, created and sold 'fake' journals to pharmaceutical companies to publish 'results' supporting their drugs, purchased the *Social Science Research Network* and created paywalls or removed legally shared working versions of articles, charge fees for open access articles, and actively lobbied against open access legislation (For a concise summary with links see Tal Yarkoni's blog entry https://www.talyarkoni.org/blog/2016/12/12/why-i-still-wont-review-for-or-publish-with-elsevier-and-think-you-shouldnt-either/). This brought massive counter movements against Elsevier in the scientific community (for example, The Cost of Knowledge). You can take action and refuse to review for or publish with unethical publishers if you feel it is justified. Thus, you should inform yourself about the publishers. Your libraries are a source of information, because they deal with the business side of publishers.

If you are in Europe, check if your institution is a signatory of ProjektDEAL. A consortium of universities is collectively bargaining with publishers via ProjektDEAL demanding that publishers reduce fees and eliminate the double paying of universities. The primary objective is that publishers sign country-wide subscription agreements that

enable access for all universities at once. Wiley agreed to such a model and this marks a paradigm change. It indicates how the publishing industry looks in the future, so long as the OS Movement proceeds. If you are not in Europe, consider starting a similar initiative, for example the entire University of California system of 10 universities, 5 medical centers, and several research institutions that collectively produce roughly 10% of the world's academic publications recently followed ProjektDEAL and boycotted Elsevier [73]. Elsevier is not the anti-science, there are many devious practices across publishers and there is an entire scientific shadow industry of predatory publishers. It requires taking action and doing your own research to take a position in favor of open science. You will not be able to be squeaky clean in this process. For example, as I became aware of Elsevier's practices around 2018, I simultaneously had an article accepted for publication in one of their journals, a very important one in the history of sociology that was founded by James S. Coleman. As a non-tenured scholar, I chose to proceed with publication rather than go through another potential two years in the review process; I am an example of the problem if not part of the problem in this case.

You can work around the publishing business. Prior to submitting an article or after it is published, you have the right to share a *preprint*—a draft of the paper you share publicly so long as it is not published elsewhere or sold for profit. Posting preprints reduces the power that publishing firms have over science, in addition to giving others immediate access to your work. However, simply posting preprints on your academic website is not open enough. Use a preprint service, for example through the *Open Science Framework*, to ensure that your preprints appear in search engines such as *Google Scholar*. *SocArXiv* for example, is the go-to location for sociology. This enables scholars to find and directly access research results based on the words they contain, uninhibited by paywalls—a crucial aspect to practicing sociology in the Global South. Preprint services are free and open access.

Practicing sociologists are generally in favor of open access. A recent Wiley author survey suggested that 67% of respondents would like to see their academic societies provide more open access publications [74]. In 2018, open access was the sixth most important thing an academic society should be doing, and in 2019, it was number one. This is a monumental shift in just one year. Changing to open access would be financially harmful to publishing houses and reduce the revenues that academic societies draw from the publication process—a key reason the ASA does not support open access requirements attached to grant funding [75]. This means we are operating in an academic setting where stockpiling money in academic associations often takes precedence over removing barriers to global access to science.

Complete decommodification of the publishing system is unlikely. It is unreasonable to assume that frustrated researchers will find more open access, non-profit journals like *Sociological Science* as this requires high quality editors willing to do an extreme amount of work without pay. As there are already many open access journals, sociologists should consider changing them from within. The MDPI publishing platform is a case-in-point. It is a for-profit publisher; however, it has done two things that suggest it is willing to work within the open science movement rather than against it. The author processing charges (APCs) are competitive. At first, they were nearly the lowest among open access publishers (345 CHF). The fees have subsequently risen to 1000 CHF (about US$ 1050), but they continue to provide steep discounts, exemptions, and fellowships for emerging scholars such as their 'Publications Travel Award'. They are also less expensive than the goliath PLoS (charging US$ 1695) and publishing over 20,000 articles per year. PLoS is a non-profit organization, and this is ideal from a normative standpoint. In addition, the developed a strong reputation in general sciences for their promotion of open science.

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
