# Peer review of "Does Sociology Need Open Science?"

_societies, doi:10.3390/soc11010009_

Round 1
Reviewer 1 Report
I consider that my main concerns indicated in the previous evaluation have been considered and are reflected in the manuscript.
Author Response
Thank you! I am happy i could meet the reviewer's expectations.
Reviewer 2 Report
See the attached!

Author Response
Thank you for the review. Unfortunately, Reviewer 3 has pushed me to decide whether I will tackle the 'why' or the 'how' question, but to do both is too much for one article. Therefore, I have focused on 'why', because i think it is important to first raise awareness before pushing for big changes. I have moved the suggestions I have about 'how' into an appendix in case readers want some ideas.
Reviewer 3 Report
Thanks for the opportunity to provide a review for the revision of your paper. I think that the manuscript has improved, and I acknowledge that you put some effort into addressing my concerns. However, I still see substantial problems.
- I still wonder what the contribution is. You now have categorized the paper as a “conceptual paper”. According to the author guidelines, based on which I need to do my review, this type of paper needs to provide “new insight into, new ways of looking at existing knowledge and concepts”. I wonder if this is the case. Do you provide a new way of looking at existing knowledge? Is a call for open science in sociology new? Can you clarify your contribution here? Perhaps one issue is that your paper has currently two main purposes. It on the one hand argues that sociology as a discipline should change, and on the other hand wants to provide a guideline for individual sociologists who want to adopt open science practices. In my opinion this two-fold approach does not really work. Maybe one way to go, is to concentrate on one of these orientations. This, of course, can mean that the one is still included in the other – but a clear choice of what the paper wants to say (call for change or instruction for individual sociologists) might help. Right now, I think that the guidelines remain too ambiguous to be actually instructive. I will come back to this further below.
- In my opinion some of the central gaps in your argumentation that I outlined in my past review have not been addressed. In response to my former second comment, you now argue that “faking data is nearly impossible”. However, this statement seems again mainly stem from your own, narrow research perspective. You acknowledge in your response that this statement is supposed to be only valid for quantitative research – but this is not mentioned in the paper. Moreover, I have done quantitative research, e.g., social network analysis. To fake data there seems really not to be a problem. If I want to assert that one actor has a tie to another, I simply type a “1” instead of a “0”. Who can detect that in anonymized data? I see your point that in some cases transparency will definitely help. But I really cannot see how transparency is supposed to be the magic means.
- Another issue that has not been addressed: Who should be in charge of deciding whether sharing data and/or materials is ethical/legal or not? You acknowledge now that data sharing might not be an option for forms of qualitative research. But you do not offer a solution for this problem, which is more than a technicality. Can the researchers themselves make this decision?
- Connected to this point: While you are more sensitive for the peculiarities of qualitative research now, you uncritically promote the TOP guidelines. In my opinion these guidelines promote clearly an ideal of science in which data sharing should be the default option while not sharing is the exception. I think this is highly problematic because these guidelines paint many qualitative research practices as abnormal. For exactly this reason, I myself have become hesitant to submit to Elsevier journals. You say that “so far the OS Movement failed to offer clear ideas and solutions about transparency for those working in qualitative research” (p. 21). But in my opinion the OS movement is treating qualitative research as abnormal as long as it uncritically supports things like the TOP guidelines. I understand that you have a different opinion there, but I think that in such a paper you need to at least critically reflect on these issues. I want to say that your suggestion on page 23 to “allow controlled access to the data” seems a way more viable solution to me. In fact, this is something I have done with sensitive data myself – I shared it with the reviewers of a paper so that they could check it.
- Also connected to this: I still do not see the point of the Riemann study. You insist that: “when several researchers look at the same data and come to very different conclusions, the knowledge gained is more about the researchers than the subject.” But I know of no qualitative researcher who would object to that (at least no one who publishes in international journals). For that reason, debates and accounts of reflexivity (reflecting on the role and impact of the researcher for the interpretation) is both immensely important and pretty much standard in qualitative research. You seem to argue against a strawman here. This is also visible in the next paragraph in which you outline recommendations for designing a qualitative study. But here again: the recommendations you outline are pretty much standard. Having that said my impression now is rather that many of the current practices in qualitative research are already in line with demands by the OS movement as you outline them, which brings up the question why the movement demands changes of “current practices”.
- My last point regarding your understanding of qualitative research practices comes back to my aforementioned issue with the ambiguity of the guideline part of your paper. On page 24 you suggest: “Regardless of the methods you use there are many opportunities to take advantage of preregistration. Some forms of qualitative research, for example those involving grounded theory and interpretivist methods, require decisions during the research process that cannot be foreseen. This uncertainty can be outlined in a preregistration stating explicitly when flexibility is and is not admissible. Moreover, simply putting a qualitative research plan online prior to conducting the research is equivalent to a pre-analysis plan. This research design need not compromise your data collection work because you can register the plan on a platform like the Open Science Framework and then embargo it, so that it is preserved but not made public until after the research concludes.”
My problem here is actually: I do not understand what you want me to do here? How would a research plan look like, e.g., for an open project with unstructured data gathering methods? Would I then just post the research question and a statement that the study is conducted through unstructured observations and interviews? - My last point comes back to revealing your identity. You do not identify yourself as an editor anymore, but you still reveal your identity. The journal guidelines state that the peer review process is supposed to be “double-blind” – so I find it puzzling that you as an editor still have not anonymized your paper.
Author Response
I cannot begin to express my gratitude to this reviewer. Through two rounds of review I have developed a new understanding of the role of open science in qualitative research. I have done my best to adapt this paper as such. I strongly urge the editors of this journal to offer this reviewer the chance to publish a comment on my paper if it is accepted eventually. I think this person has a lot to add, and this action would demonstrate good open science principles on the part of the journal. Moreover, this reviewer has invested so much time into this process that they should get to have a public voice, a chance to obtain 'credit' for all this hard work. I also think their perspective is valid and would contribute to science if presented.
Here are my responses to the specific points raised by the reviewer.
- What is the contribution? I have now added a section right at the beginning of the paper called "Open Science: Some Housekeeping". In this section I reveal the purpose of the paper as a primer or overview on the state of sociology and why practices should change. I now admit that the paper is just my own perspective and point out that the paper is successful if it empowers both supporters and opponents of open science. I have removed the section offering practical advice on how to do open science and put in in Appendix E. I have also added Appendix D which quickly covers some diverse (including feminist) definitions of open science.
2. I have backed off from my position that faking data is nearly impossible and simply suggest now that transparency would reduce this practice. The network data is a good point, but I believe there are methods that can identify if things are truly random or not, and if data were altered that scholars might find a way (like observing the occurence of zeros in the third digits of the Steward data!). But I suppose this is for a different article.
3. Who decides what is ethical: I have added a paragraph in the conclusion pointing out that this question can lead to divisions and power struggles and that this paper has a more humble goal of raising awareness. I have also added a paragraph at the bottom of p. 23 that points out the value in discussing research designs prior to conducting research to achieve greater ethics, without taking a stance on who enforces ethics. Again, the paper now has a more humble goal to spark discussion, for example what has taken place with the reviewer here. Another reason why the journal should offer a chance to publish a comment would be very transparent and helpful, especially as this person can clearly articulate positions of qualitative researchers that I cannot.
4. TOP guidelines. I agree that any guidelines are problematic that would make qualitative methods 'abnormal' and I have added new text to the section "Resistance..." on p. 23 that discuss the transparency suggestion in the TOP guidelines and i take a strong stance against any guidelines that would render qualitative methods unethical or abnormal.
5. I think I see your point about the Riemann study. I now try to frame this in a positive way. The study is evidence of the subjectivity of researchers which is not a bad thing, but we need to be careful about interpreting the data out of context. i have removed the paragraph outlining suggestions for how to design a qualitative study. This was not helpful and more reflected my own ignorance. See more details below.
6. In the specific example of grounded theory, yes. i would say that it would be very useful to preregister a grounded theory study. The researcher would still need to provide evidence of their target population and why they thought they could learn something from that population. If subjectivity is to be embraced, then motives should be revealed.
Related to several points, especially 4-6, I have rewritten key references to qualitative research. Most of p. 22 now discusses that qualitative methods (and every method really) deserve their own transparency considerations. I now point out how many qualitative researchers already practice open science. However, to counterbalance this, I also now cite a study that demonstrates transparency problems in qualitative research on p. 12.
Round 2
Reviewer 3 Report
I am genuinely and positively surprised about the significant progress of this paper throughout the revisions. I did not expect this. I acknowledge that you spent a lot of effort into revising this paper and I think this paid off. I still have some strong reservations about several aspects, but I see no substantial flaws in argumentation anymore. You have other positions than me, but these are argued well enough now, so I see no issues left that should prevent publication.
I see all these ethical problems that you identify, and I have seen a lot unethical research practices first-hand – and I think that you only present the tip of the iceberg here. I simply do not think that Open Science will solve most of these problems. My impression is that those colleagues who conduct unethical practices seem to be those who have mastered being opportunistic to all kinds of expectations from their peers. I am afraid that they will simply adapt while those “naïve” scholars who are just trying to do good sociology might struggle the most with open science demands. However, this personal opinion of mine should not be a reason to give your paper a bad assessment.
The most important improvement, which in my opinion now sells the paper, is your reflexivity. The paper is now way more honest and clear about its motives and intentions. To me this degree of reflexivity is one of the most important aspects of research – which in itself maybe could also be considered an “open science” practice.
I only have some minor suggestions that you might or might not want to consider before possible publication.
- You should check the English again. I am not a Native Speaker but the sentence “That science in practice is problematic” seems a bit strange. What does this mean?
- On page 2, I would at least briefly acknowledge the possibility that sociology might be different from other disciplines. Perhaps sociologists are more ethical, perhaps not. My own personal experience suggests the former – but this is of course only anecdotal. I have done interdisciplinary projects and have come across unethical practices – but interestingly mostly conducted by business and educational scientist, not so much by sociologists. There also some other indicators – like the non-existence of journal rankings in sociology(!) in contrast to other disciplines – that probably affect research practices and are a result of these. I think you can easily acknowledge that sociology might be different and nevertheless argue that it is plausible that sociology is not significantly different or at least similarly unethical. In doing so, you run a lower risk of losing readers at this point.
- I would drop the following sentence: “and in a US survey, only 20% 106 of Americans believed that scientists always acted in the public interest [20].” (page 3, lines 106-107). This does not support your argument because, of course, scientists do not always act in the public interest. Which scientist would actually say otherwise? I certainly do not always act in the public interest because I unsurprisingly have own interests as well.
- Page 4, lines 161f read: “To delineate their work as not sociology, is not open science. It would construct a disciplinary wall…” I would drop the “is not open science”, because this does not make sense to me here. I would just write: “To delineate their work as not sociology would construct a disciplinary wall…”, since these seems to be the actual argument here, right?
- On page 11, lines 473-475 you write: “When researchers do not plan to share data by default, the problem is procedural rather than simply a reflection of researcher preferences.” Do you want to suggest that planning not to share data by default is a “problem”? Because it reads this way and I think this would be a problem. There are many good reasons why you would plan not to share data by default. Just an example: We did so in a project “by default” because it allowed for better access to our research field. If I would not have assured our gatekeeper that we would not share data, they would not have let me observe confidential meetings.
Author Response
Thanks again to Reviewer 3. The time invested in this paper will benefit me, this journal and most importantly open science. At least in my opinion.
I have adjusted the awkward sentence reading "that science in practice is problematic" to be more clear now.
I added a few things on p. 2 to suggest that psychology might be uniquely unethical and that sociology might be different. I then state "I assume" and "If nothing else, it is better to err on the side of caution when trying to study human societies."
I think scientists when doing "science" should be acting in the public interest, or their own interests should be the public interest, which should be accurate science. I have adjusted this section to be less misleading now and more clear, that this is an image problem more than anything else.
I fixed the "delineate" sentence now, thanks!
I added a sentence about certain studies that need trust and data protection and: "If I were speaking on behalf of the open science movement I would say this is no problem, we should only share data in all cases when it is possible."
This manuscript is a resubmission of an earlier submission. The following is a list of the peer review reports and author responses from that submission.
Round 1
Reviewer 1 Report
Thank you for the possibility to read this manuscript.
This "Communication" is extremely interesting and pertinent and somewhat discutable (no problem) in today's world in which open science could be a central dimension of the scientific knowledge legitimation.Some comments/suggestions:
- the number of pages is 22 and not "33";
- Needs clarification: "Reliability, transparency and ethical crises pushed psychology to reorganize as a discipline over the last decade. Political science also shows signs of reworking itself in response to these crises. Sociology sits on the sidelines. There have not been the same reliability or ethical 6 scandals, at least not in the limelight, nor has there been strong disciplinary moves toward open science." (Abstract);
- there are many keywords but it seems to me that "sociology legitimization"could be one of them (not mandatory);
- edit: "This is a problem I believe needs to be addressed immediately.";
- justify with more references "Scandals that brought social science";
- I suggest that the footnotes be moved to the primary text;
- this subtitle seems to me to be susceptible to confusion so I suggest it be reformulated: "2.2. Open Science is Sociology";
- reveals an excellent knowledge of Weber, Merton and Habermas;
- Very pertinent and well justified: "3. Resistance to Open Science";
- about "4.1. Transparency", please clarify "Make all the materials associated with a research paper or book available online." and how "Transparency is not that much extra work."?.I think the paper seems too optimistic in: "4. Actions to Take Now." chapter Please, check;
- In "5. Conclusion: The Theory in Open Science", be more specific about the answer about the Communication objective: Sociology needs open science?
In a word: the manuscript has many potential but seems necessary some work for a better "defense" of its proposal.
Author Response
thank you for the comments.
I have reworded the abstract entirely and somewhat the introductory paragraph so that it is not as confusing hopefully.
Thank you for the "sociology legitimation" keyword. i have also used this phrase in some places in the new version of the paper.
Like Reviewer 3 commented, I have removed the part about it being 'not too much work'. I was wrong. I have advocated for sharing materials, but only in certain circumstances.
I like the title Open Science is Sociology, but I feel that it could be confusing because open science is for every discipline. But I also don't like "Sociology is Open Science", so I guess I will keep the title for now. I hope that is fine.
Reviewer 2 Report
This essay aims to present why and how sociology as a science should take a sort of paradigm shift toward open science. The author(s) focuses in particular on the functional nature of the ordinary sociology in terms of its methodological ethics, arguing that sociology has recently begun to shift its methodological tenor to "open" science, which it can, and should, be more transparent, reproducible, accessible, and less biased. Although the author(s), depending on some key conceptual ideas from Weber, Merton and Habermas, deal shortly with the polemical notion of "value-neutral" in the domain of social sciences, their questioning, suggestions and arguments appear to have provocative implications for the debates on the nature and practical function of sociology as a science.
Overall, the essay is of focus and coherence, in the sense that the development of its theme maintains logical consequences consistently.
There is one request the reviewer would like to recommend. The length of the sentence in the conclusion section is too short to reflect the key points discussed and suggested by the author(s) in the main discussion part of the essay. So the author(s) are encourged to increase the length of the conclusion section by adding at least one more paragraph.
Author Response
Thank you for the supportive words. Regarding the conclusion. I think there was a misunderstanding. The conclusion is only a final point. It is not a summary of the entire paper. I wanted to make one last point about theory being important. The paper otherwise does not need a conclusion in my opinion because each section stands alone and is useful, and the big open science conclusion came before the section on Actions to Take Now. I hope that is fine.
Reviewer 3 Report
Thank you for the opportunity to review the manuscript “Does Sociology Need Open Science?”. The paper asserts that sociology needs to adopt open science practices in the form of sharing all research materials, provide open access to all data, preregistering research plans, and more. It argues that such practices should be enforced through the use of sanctions employed by, for instance, research associations and institutions. The manuscript apparently is supposed to be some kind of opinion piece or viewpoint, but in fact turns out to be an activist manifest – and in no way a scientific paper. The argumentation is mostly inconclusive, partly even contradictory in itself, and evidence and empirical examples are selected with obvious bias. In section 4 the paper offers some interesting suggestions of how to improve credibility in the research process. However, calls for “transparency” or “decommodification of science” are hardly new and I see no suggestions that would add anything to the existing discourse. Moreover, the suggestions mostly apply only to quantitative research such as the idea of “pre-registered” research. The overall call for open science in sociology is not derived from the argumentation; instead the arguments appear erratically selected to support the pre-defined political position. Perhaps it is suitable for an activist blog or so – but it is not a scientific paper. To explain my impression, I will try to outline some of the major issues in the following:
- A paper that is throwing such serious accusations about questionable research practices should be very careful when it comes to arguing in such a bold way without substance. The paper contains many bold and controversial statements without grounding these in solid evidence or argumentation. For example, there are assertions such as: “It does not matter if the research is qualitative or quantitative. Researchers could easily only report those parts of texts, interviews and observations that support the theory they want or the interpretations they prefer.” (p. 2) Statements such as this are not backed by anything and are not plausible at all. You clearly do not know about the relevance of reflexivity and plausibility in qualitative research and how this is a crucial issue in the review process. You seem to simply transfer anecdotal experiences from quantitative research review processes to qualitative research with no knowledge of how things actually work there.
Another statement is: “Sociology journals, for example, do not retract articles despite evidence of serious methodological mistakes.” To support this claim, the paper presents some very few examples of questionable critiques of existing journal publications. Strikingly, you do exactly what you criticize in your other statement, i.e. you only report those parts of text that support the interpretation you prefer. I want to stress that these are only two examples of several in which bold accusations are not backed by solid evidence or argumentation. - The whole claim of the paper is a pretty naïve idea of the positive effects of openness without any reflection on the negative implications. For once, you seem to assume that openness would kind of magically solve problems with faking data or analyses. However, why would that be the case? Researchers can still fake their data or their data analyses. This does not change only because the fake data is made publicly available. Why would it?
Moreover, opening science to a broader public also comes with significant negative implications (not only positive). Just think of the recent media coverage of coronavirus-related research made openly available on preprint servers. As a consequence, mass media started to read and criticize the preliminary research results. Scientists suddenly faced massive public critiques for controversial or non-conformist research and were sometimes even discredited by yellow press media. Controversies between scientists were dragged into the mass media, and so on. However, when it comes to such negative implications you remain completely silent, which points to a severe lack of reflexivity in respect to your assertions. - Another naïve belief seems to be that anonymity can be easily ensured in all kinds of research (p. 13). This is not the case, neither in qualitative nor quantitative research. For quantitative as well as for qualitative data, the increasing possibilities and capabilities of Big Data analytics present a severe issue since such analyses can find patterns in data and cross-referenced data that the researcher cannot see or anticipate. “Re-Identification” is a serious problem that you apparently chose to simply ignore in your argumentation.
Moreover, there are issues unique to qualitative research. Take for example, extensive narrative interviews in which people give detailed insights into their lives. You can also think of ethnographic research in which researchers produce extensive observational protocols revealing a lot about what people do, when, where, and how. Simply redacting their names or even more does not ensure anonymity here. In fact, the only way of ensuring anonymity in such research is to carefully select certain aspects from the interviews or observations and only make these public – so exactly what you criticize as wrong. In a great many cases, the anonymization must even go so far as to actively semi-fictionalize data. Do you intend to publish (semi-)fictionalized data or do you imply that such research should simply be abandoned? What about data gathered in covert research? In such research you can, by default, not ask for participants’ consent.
As a result, a major effect of your claims would be the complete annihilation of large parts of qualitative research practices per se. A policy that enforces the sharing of all research materials and data, and the pre-registration of research plans, would virtually make all kinds of covert, semi-covert, and semi-fictional research instantaneously impossible – in addition to a lot of other qualitative research that involves narrative interviews and observations. You do not discuss any of these issues – probably because of a lack of knowledge about qualitative research practices. You might want to argue that you state that “sharing information that would jeopardize the well-being of participants is wrong” (p. 13, lines 438-439). But this argument is not thought through. How would this work in practice? Who gets to decide which parts of data can be shared and which cannot? If the researcher gets to decide it, you contradict your assertion that the researcher should not be the one who may decide if data is shared or not (p. 12). So, what would be the idea here? Unfortunately, again, you do not offer an argument to substantiate your claims. - You seem to believe that theoretical and methodological plurality are a problem, whereas it is a core principle of all sociological research grounded in a constructivist paradigm. You seem to have a problem with the fact that different sociologists use different methods and come to different conclusions even when using the same data (p. 13). But in constructivist research this would be the intended(!) effect, not a fault. The so derived results are not “random” (p. 13), they are based on argumentation and differing constructions of the world. Different methods are not selected randomly. They offer different views and interpretations on purpose. Therefore, in this respect your argumentation implies that sociology as a whole should first of all adopt a positivist research paradigm subscribing to the idea that there is some kind of one true interpretation – which is an absolutely presumptuous assertion.
You kind of counter this critique on page 13, lines 461-465 when you argue against the assertion by other scholars that transparency policies would impose a positivist understanding of science. However, your counterargument is that this is not the “position” of the Open Science movement, which is completely missing the point. The imposition of a positivist paradigm might not be a “position” of the movement, but it would the effect(!) of the demanded policy. - From chapter 3 onwards, the paper becomes strangely political. The paper extensively outlines the divide between the German Sociological Association and the German Academy for Sociology mainly taking the position of the Academy and arguing against the positions of the Association. It even starts to outline positions and membership structures and the controversies between both organizations. Why is this in the paper? Why should the reader of an international general-interest sociology journal care about the positions of a small split-off “academy” in Germany? I hardly see what this adds to the argumentation of the paper.
- Last but not least I want to point out that I consider it absolutely unethical that you casually mention that you are a member of the journal’s editorial board while you also assert in the very same paragraph that you decided to “publish” (not “submit”) “this paper in this journal” (p. 16). In my opinion this is an unacceptable attempt to influence my decision as a reviewer.
There are many other issues (like the erratic selection of three classic authors and the even more erratic selection of aspects of their theories), but I wanted to point out the most crucial ones here. As I mentioned above, I do not think that this qualifies as a scientific paper.
Author Response
This review was incredibly insightful and humbling. I was unaware of how my paper 'sounded' to others and found many of the criticisms well stated. I have made some important changes in response to this reviewers helpful criticism. I realize the reviewer wants to remain anonymous, but just know that you made this communication far better than it was before - and yes it is a "communication", not an empirical paper, as you were concerned.
- Yes, I have simplified and overstated what qualitative research is. For this reason I have made major changes in many sections. I have taken away an insistence that qualitative researchers must share their data. I have advocated for it when possible. Instead I've shifted the focus to research design and distinguishing forms of research that are either grounded theory or intentionally not designed to be 'science' or might reject such labelling or 'scientific methods'. In cases beyond these types of research I advocate for preregistering research designs. I have elaborated more on Lamont's workshop's promotion of linking previous theory and linking findings with data as key goals of qualitative research design (!! especially design). I hope this makes the arguments more attractive and friendly toward researchers using qualitative methods.
- As I mention at one point in the paper in reference to Pickett's study. Faking data is almost impossible. At least quantitative data. The algorithm's used to fake data produce results that are unnatural in many ways. So yes, in the case of quantitative research, replication will identify and eliminate faked data in almost all cases.
- As I stated in point one, i now agree that there are so many possibilities for unforeseen problems with anonymity and have removed all cases of pushing for this. Thank you for pointing this out.
- When discussing the Riemann Qualitative Sozialforschung issue, I now highlight the benefits of plurality and how important differences of interpretation are to sociology. I only use this to point out how research design is important. Again, thank you for this critique. I completely agree with your position.
- I also agree that it is not helpful to discuss the DGS and AS split in Germany and have removed it completely from the paper.
- I mention I am a member of the editorial board because I wanted to be critical of MDPI, the publisher of this journal, and to give a real life example of trying to advocate for change from within something that already exists that might prove to be a good open access publication outlet with support from open science advocates. I stated "chose to publish" in my paper because I was taught to write papers in a way that they should appear in the journal if published. Also, this is a "communication" not an original research paper, so it seems like a good idea to mention this.
Again, thank you so much for this review. It was much needed as it is my intention or vision for open science to be all inclusive and a uniting event, not one that ostracizes.